# Toward Spatial Intelligence: A Unified Self-supervised Framework for 3D Representation Learning from Unposed Multi-View Images

## Abstract

Robust 3D representation learning forms the perceptual foundation of spatial intelligence, enabling downstream tasks in scene understanding and embodied AI. However, learning such representations directly from unposed multi-view images remains challenging. Recent self-supervised methods attempt to unify geometry, appearance, and semantics in a feed-forward manner, but they often suffer from weak geometry induction, limited appearance detail, and inconsistencies between geometry and semantics. We introduce *UniSplat*, a self-supervised framework designed to address these limitations through three complementary components. First, we propose a *dual-masking strategy* that strengthens geometry induction in the encoder. By masking both encoder and decoder tokens, and targeting decoder masks toward geometry-rich regions, the model is forced to infer structural information from incomplete visual cues, yielding geometry-aware representations even under unposed inputs. Second, we develop a *coarse-to-fine Gaussian splatting strategy* that enhances appearance learning by progressively refining the radiance field, thereby enhancing appearance detail to produce high-fidelity representations. Finally, to enforce geometric–semantic consistency, we introduce a *pose-conditioned recalibration mechanism* that interrelates the outputs of multiple heads by reprojecting predicted 3D point and semantic maps into the image plane using estimated camera parameters, and aligning them with corresponding RGB and semantic predictions to ensure cross-task consistency and resolving geometry–semantic mismatches. Together, these components yield unified 3D representations that are robust to unposed, sparse-view inputs and generalize across diverse tasks, laying a perceptual foundation for spatial intelligence.

## 1 Introduction

Spatial intelligence, the ability to construct and reason over structured representations of the physical world, is a key prerequisite for embodied agents that must navigate, manipulate, and plan in complex environments. A fundamental prerequisite for such intelligence is robust 3D perception, which enables agents to build structured representations of the world that integrate geometry, appearance, and semantics. These representations serve as the perceptual foundation for downstream tasks in embodied AI, including navigation, manipulation, and scene understanding. However, learning effective 3D representations directly from unposed multi-view images remains an open challenge.

Research on 3D perception has long benefited from supervised feed-forward reconstruction methods, which aim to infer geometry, appearance, and semantics directly from images with the aid of ground-truth supervision. Neural Radiance Fields (NeRFs) (Mildenhall et al., 2021) and their extensions such as Mip-NeRF (Barron et al., 2021) and Instant-NGP (Müller et al., 2022) achieve high-fidelity novel-view synthesis but require calibrated multi-view images and often rely on per-scene optimization. The introduction of 3D Gaussian Splatting (Kerbl et al., 2023) accelerated training and rendering by representing appearance with explicit primitives, and subsequent variants have explored pose-aware (Chen et al., 2021; Xu et al., 2024a; Chen et al., 2024a; Tang et al., 2024) and pose-free pipelines (Jiang et al., 2024; Wang et al., 2024a; Smart et al., 2024; Ye et al., 2025). In parallel, semantic scene fields (Zhi et al., 2021; Peng et al., 2021; Fan et al., 2024; Li et al., 2025a) have made progress toward 3D semantic understanding, while geometry-focused works predict cam-

era poses or dense point maps to recover scene structure (Wang et al., 2024b; 2025; Zhang et al., 2025; Leroy et al., 2025). Despite this progress, most supervised methods depend on ground-truth geometry or calibration signals and tend to treat geometry, appearance, and semantics in isolation, leaving a gap for unified perceptual representations that support spatial intelligence.

To reduce dependence on costly 3D labels, self-supervised approaches aim to build geometry- and view-aware priors from unlabeled images. Inspired by 2D representation learning, masked autoencoding (He et al., 2022; Bao et al., 2022; Dong et al., 2025) and contrastive objectives (Chen et al., 2020b; Grill et al., 2020) have been extended to 3D, promoting cross-view invariance and reconstruction consistency (Weinzaepfel et al., 2022; Zhu et al., 2023b). Novel-view synthesis has also been widely used as a training signal, although many methods assume dense video supervision and degrade in sparse-view regimes (Bian et al., 2023; Fu et al., 2024). Recent work moves toward pose-free self-supervision by jointly estimating cameras and scene structure (Jiang et al., 2025a; Kang et al., 2025), highlighting the importance of anchoring representations in a consistent spatial frame. Meanwhile, feed-forward Gaussian-splatting models such as UniForward (Tian et al., 2025) and Uni3R (Sun et al., 2025) attempt to unify geometry, semantics, and appearance in a single pipeline. Yet most of these methods still suffer from weak geometry induction, limited appearance detail, and geometry–semantic inconsistencies. This motivates the development of unified frameworks that explicitly couple geometry, appearance, semantics, and camera estimation to form a consistent perceptual basis for embodied AI.

To address these challenges, we propose *UniSplat*, a self-supervised framework for learning unified 3D representations from unposed multi-view images. UniSplat is built on a transformer encoder with a multi-head decoder and introduces three key innovations. First, a *dual-masking strategy* enforces geometry-aware feature learning by masking both encoder and decoder tokens, with decoder masks targeted to geometry-rich regions, thereby strengthening geometry induction from incomplete visual evidence. Second, a *coarse-to-fine Gaussian splatting strategy* hierarchically refines the radiance field, progressively enhancing appearance detail to produce high-fidelity visual representations. Finally, a *pose-conditioned recalibration mechanism* enforce geometric–semantic consistency by interrelating decoder predictions. Unlike conventional multi-task learning, where each head operates independently, our design uses estimated camera poses to reproject 3D point and semantic maps into the 2D image plane and align them with the corresponding RGB and semantic predictions, ensuring cross-task coherence and resolving geometry–semantic mismatches.

Our contributions can be summarized as follows:

- We introduce a dual-masking strategy that applies masking to both encoder and decoder tokens, with decoder masks biased toward geometry-rich regions to encourage geometry-aware representations from incomplete cues.
- We propose a coarse-to-fine Gaussian splatting strategy that hierarchically refines the radiance field, enhancing appearance detail and producing high-fidelity visual representations.
- We design a pose-conditioned recalibration mechanism that reprojects 3D point and semantic maps into the image plane using estimated camera poses and aligns them with RGB and semantic predictions, enforcing cross-task coherence.

Together, these contributions enable UniSplat to learn unified 3D representations that are robust to unposed, sparse-view inputs and broadly transferable across tasks, laying a perceptual foundation for spatial intelligence. Experiments on diverse 3D scene understanding and embodied AI benchmarks confirm consistent performance gains, validating the generalization ability of our framework.

## 2 RELATED WORK

### 2.1 SUPERVISED 3D REPRESENTATION LEARNING

Supervised feed-forward approaches aim to recover geometry, appearance, and semantics in a single pass using explicit supervision such as target-view rendering signals or external priors. These methods allow fast inference without per-scene optimization (Kerbl et al., 2023; Lu et al., 2024; Qin et al., 2024; Zhou et al., 2024). A key distinction lies in their dependence on camera poses. **Pose-required models** assume known intrinsic and extrinsic characteristics during both training and testing. They often rely on epipolar constraints, cost volumes, or pose-conditioned embeddings, achieving strong photometric quality (Chen et al., 2021; Xu et al., 2024a; Charatan et al., 2024; Chen et al., 2024a;

Tang et al., 2024; Xu et al., 2024c; Zhang et al., 2024; Xu et al., 2024b). However, these methods depend heavily on SfM-style preprocessing and may fail when pose estimation is unreliable.

**Pose-free models** remove pose inputs at inference but still require posed supervision or fixed targets during training. Within this family, research has diverged by task emphasis. Geometry-focused models predict camera parameters or dense point maps aligned with the scene structure (Wang et al., 2024b; 2025; Zhang et al., 2025; Leroy et al., 2025; Jiang et al., 2025b; Li et al., 2025b). Appearance-focused methods directly predict per-pixel Gaussians in a canonical space, resolving scale by encoding intrinsics (Jiang et al., 2024; Wang et al., 2024a; Smart et al., 2024; Ye et al., 2025). Semantics-focused approaches lift 2D vision-language features into a 3D-consistent field, enabling open-vocabulary, view-consistent segmentation (Fan et al., 2024; Li et al., 2025a; Sheng et al., 2025). While these strands are beginning to converge, the reliance on posed training combined with unposed inference can lead to residual inconsistencies, motivating the development of self-supervised alternatives.

## 2.2 SELF-SUPERVISED 3D REPRESENTATION LEARNING

Self-supervised approaches aim to reduce the reliance on costly 3D labels by learning 3D representations directly from raw multi-view images. **Early methods extended ideas from 2D representation learning**: contrastive learning encouraged view-invariant features (Chen et al., 2020b; Grill et al., 2020), while masked autoencoding and cross-view completion promoted reconstruction and correspondence (He et al., 2022; Bao et al., 2022; Weinzaepfel et al., 2022; Zhu et al., 2023b; 2025; Dong et al., 2025). These methods improved feature learning, but they often lacked strict global 3D consistency and produced representations that were more view-aligned than spatially grounded. Another line of work employed **novel view synthesis** as a self-supervisory signal, where models were trained to render unseen target views and match them photometrically (Bian et al., 2023; Fu et al., 2024). Although this supervision tied predictions more directly to 3D structure and improved geometry–appearance coupling, most approaches assumed known or pre-estimated camera poses or leveraged video metadata to simplify correspondence. Moreover, they typically require dense video streams and re-render nearby frames, which limits robustness in sparse-view settings and constrains applicability to real-world scenarios.

More recently, **pose-free self-supervised methods** seeks to remove this dependency by jointly estimating cameras and scenes directly from raw, unposed image collections. RayZer (Jiang et al., 2025a) exemplifies this direction with a transformer-based latent renderer that couples camera and scene recovery in a predict-then-render loop. SelfSplat (Kang et al., 2025) employs explicit Gaussian splatting, predicting depth and pose with separate modules, which yields interpretable outputs but less coherent alignment. Latent models offer flexibility, while explicit ones provide interpretability and efficient rendering; both highlight a shift toward joint camera–scene learning. Parallel efforts such as UniForward (Tian et al., 2025) and Uni3R (Sun et al., 2025) attempt to unify geometry, semantics, and appearance in a single feed-forward pipeline under self-supervision. Despite this progress, current pose-free methods still suffer from weak geometry induction, limited appearance detail, and geometry–semantic inconsistencies, underscoring the need for frameworks that explicitly couple all three aspects with camera estimation in a consistent 3D reference frame. In this paper, we propose UniSplat, a unified self-supervised framework that strengthens geometry induction through dual-masking, enhances appearance fidelity via coarse-to-fine Gaussian splatting, and enforces geometry–semantic consistency with pose-conditioned recalibration, yielding robust 3D representations that transfer effectively to scene understanding and embodied AI tasks.

## 3 METHODOLOGY

### 3.1 OVERVIEW

We propose UniSplat, a unified feed-forward framework for self-supervised 3D representation learning from unposed multi-view images. As shown in Figure 1, the model consists of a transformer encoder and a multi-head decoder that predicts dense point maps, semantic maps, RGB renderings, and camera parameters. A dual-masking strategy (§3.2) strengthens geometry induction by masking both encoder and decoder tokens, with decoder masks biased toward geometry-rich regions. A coarse-to-fine Gaussian splatting strategy (§3.3) progressively refines the radiance field from global structure to semantic context and fine appearance, enhancing visual detail. Finally, a pose-conditioned recalibration mechanism (§3.4) reprojects predicted 3D point and semantic maps

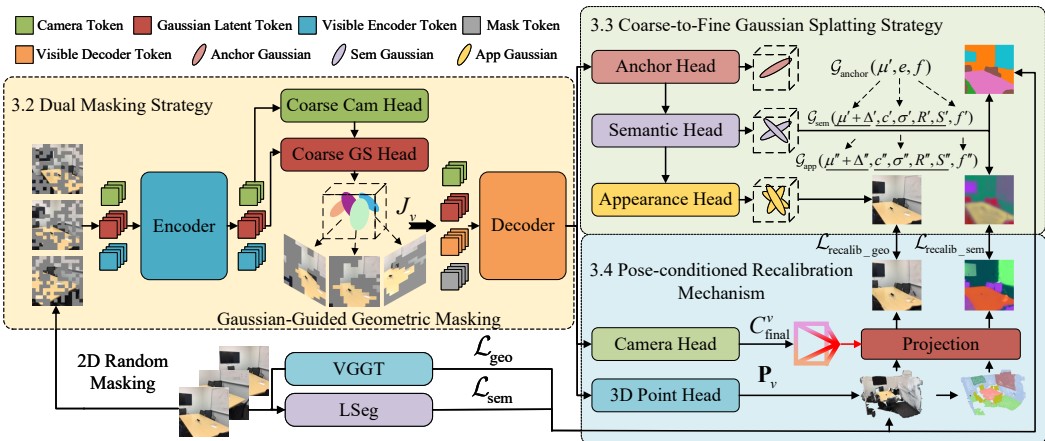

Figure 1: **Overview of the proposed UniSplat framework.** UniSplat integrates a dual-masking strategy for geometry induction, a coarse-to-fine Gaussian splatting strategy for appearance refinement, and a pose-conditioned recalibration mechanism for geometry–semantic consistency. See §3.

into the image plane using estimated camera parameters and aligns them with RGB and semantic predictions, ensuring cross-task consistency. The training objectives are introduced in §3.5.

## 3.2 DUAL MASKING STRATEGY

To strengthen geometry induction, UniSplat employs a **dual-masking strategy** inspired by (Wang et al., 2023) that applies masking to both encoder and decoder tokens. By targeting decoder masks toward geometry-rich regions, the model is encouraged to infer global 3D structure from incomplete visual cues rather than overfitting to trivial textures.

**Stage 1: Initial Masking and Augmented Encoding.** Given a set of multi-view images $\mathbf{I} = \{I_v\}_{v=1}^V$, we partition each into patch tokens $\mathbf{X} = \{X_v\}_{v=1}^V$, where $V$ is the number of views. A random mask $\mathbb{M}_{e,v}(\rho_e)$ with ratio $\rho_e$ is applied to each $X_v$:

$$X_v^{\text{vis}} = (1 - \mathbb{M}_{e,v}) \odot X_v, \tag{1}$$

yielding the visible set $\mathbf{X}_{\text{vis}} = \{X_v^{\text{vis}}\}_{v=1}^V$. Following (Jiang et al., 2025a), the encoder $E_\theta$ is augmented with learnable camera tokens $T_{\text{cam}} \in \mathbb{R}^{V \times d}$ and Gaussian latent tokens $T_{\text{coarse}} \in \mathbb{R}^{N \times d}$, where $N$ is the number of latent Gaussians, and $d$ is the latent dimension. The encoder processes the concatenated token sequence $[X_v^{\text{vis}}, T_v^{\text{cam}}, T_{\text{coarse}}]$ as follows:

$$[\mathbf{Y}_{\text{vis}}, T'_{\text{cam}}, T'_{\text{coarse}}] = E_\theta([\mathbf{X}_{\text{vis}}, T_{\text{cam}}, T_{\text{coarse}}]), \tag{2}$$

where $\mathbf{Y}_{\text{vis}} = \{Y_v^{\text{vis}}\}_{v=1}^V$ are encoded features, $T'_{\text{cam}}$ are updated camera tokens, and $T'_{\text{coarse}}$ are updated Gaussian latent tokens.

**Stage 2: Gaussian-Guided Geometric Masking.** Next, encoded features are used to guide a second geometry-aware masking pass. Updated camera tokens $T'_{\text{cam}}$ are passed to a *Coarse Camera Head*:

$$C_{\text{coarse}} = H_{\text{cam}}^{\text{coarse}}(T'_{\text{cam}}), \tag{3}$$

where $C_{\text{coarse}} = \{c_v \in \mathbb{R}^9\}_{v=1}^V$ encoding the intrinsics and extrinsics of images. In parallel, Gaussian tokens $T'_{\text{coarse}}$ together with $C_{\text{coarse}}$ are passed to a *Coarse Gaussian Head*, forming a preliminary geometric Gaussian field $\mathcal{G}_{\text{geo}}$:

$$\mathcal{G}_{\text{geo}}(\mu, \sigma, r, s, \beta) = H_{\text{gauss}}^{\text{coarse}}(T'_{\text{coarse}}, C_{\text{coarse}}), \tag{4}$$

where each Gaussian has the center position $\mu_k \in \mathbb{R}^3$, opacity $\sigma_k \in \mathbb{R}^+$, rotation $R_k \in \mathbb{R}^4$, scale $s_k \in \mathbb{R}^3$, and learnable importance score $\beta_k \in \mathbb{R}^+$. To identify the most structurally critical regions, we render a geometric importance map $J_v \in \mathbb{R}^{H \times W}$ via alpha blending:

$$J_v = \sum_{i=1}^N \sigma_i \beta_i \prod_{j=1}^{i-1} (1 - \sigma_j). \tag{5}$$

For each patch, an average importance score is obtained by pooling pixel values from $J_v$. Patches exceeding the dual masking threshold $\rho_d$ are selected, defining a geometry-aware mask $\mathbb{M}_{d,v}(\rho_d)$. Applying this second mask to the visible tokens from the first stage yields:

$$Z_v^{\text{vis}} = (1 - \mathbb{M}_{d,v}) \odot Y_v^{\text{vis}}. \tag{6}$$

The resulted $\mathbf{Z}_{\text{vis}} = \{Z_v^{\text{vis}}\}_{v=1}^V$ will be passed to the decoder.

By selectively hiding structurally important features, this dual masking forces the decoder to reconstruct them from sparse evidence by reasoning about the underlying 3D spatial structure rather than relying on local texture completion, thereby improving geometry-aware representation learning.

### 3.3 COARSE-TO-FINE GAUSSIAN SPLATTING STRATEGY

A central challenge in unified 3D representation learning is the granularity mismatch between semantic and appearance representations: semantic fields are coarse by nature, while appearance fields require dense, fine-grained primitives to capture textures and lighting. To reconcile this and enhance appearance learning, UniSplat introduces a **coarse-to-fine Gaussian splatting strategy** that progressively refines scene representations from global structure to fine detail.

The decoder takes as input the visible tokens $\mathbf{Z}_{\text{vis}}$, updated camera tokens $T'_{\text{cam}}$, Gaussian latent tokens $T'_{\text{coarse}}$, and learnable masked tokens $T_{\text{mask}}$, and predicts multiple scene properties through a **multi-head design** inspired by (Wang et al., 2025). Specifically, the *Point Head* uses a Dense Prediction Transformer (DPT) to regress per-view 3D point maps $\mathbf{P}_v \in \mathbb{R}^{H \times W \times 3}$, the *Camera Head* refines camera parameters $C_{\text{final}}$ from $T'_{\text{cam}}$, the *Gaussian Head* predicts physical attributes of 3D Gaussians (center, color, scale, rotation, opacity) for appearance modeling, and the *Semantic Head* predicts semantic features for each Gaussian to support scene understanding.

Building on these outputs, our coarse-to-fine Gaussian splatting proceeds in three stages. First, the *Anchor Gaussian Head* predicts anchor Gaussians $\mathcal{G}_{\text{anchor}}(\mu', e, f)$ from latent tokens $T''_{\text{coarse}}$, where $\mu'$ denotes center position, $e$ is the geometric feature, and $f$ is the semantic feature. Similar to Scaffold-GS (Lu et al., 2024), each anchor Gaussian serves as a base from which multiple semantic Gaussians are derived by applying learned position offsets, enabling richer coverage of the local scene context. Next, the *Semantic Gaussian Head* expands each anchor into semantic Gaussians $\mathcal{G}_{\text{sem}}(\mu' + \Delta', c', \sigma', R', S', f')$ by predicting offsets $\Delta'$, coarse appearance attributes $(c', \sigma', R', S')$, and semantic features $f'$. These semantic Gaussians can be rasterized into 2D maps through alpha compositing:

$$s = \sum_{i=1}^{N_s} \sigma'_i f'_i \prod_{j=1}^{i-1} (1 - \sigma'_j), \tag{7}$$

where $N_s$ is the number of semantic Gaussians. Finally, each semantic Gaussian acts as a new anchor to diffuse a denser set of fine-grained appearance Gaussians $\mathcal{G}_{\text{app}}$, with refined attributes $(c'', \sigma'', R'', S'')$ predicted by the *Appearance Gaussian Head*. The whole process can be summarized as:

$$\mathcal{G}_{\text{anchor}}(\mu', e, f) \Rightarrow \mathcal{G}_{\text{sem}}(\mu' + \Delta', c', \sigma', R', S', f') \Rightarrow \mathcal{G}_{\text{app}}(\mu'' + \Delta'', c'', \sigma'', R'', S'', f''). \tag{8}$$

By progressively refining anchor, semantic, and appearance Gaussians, this strategy resolves the granularity mismatch between coarse semantics and fine textures, producing 3D representations that enable rendered outputs to capture fine appearance detail.

### 3.4 POSE-CONDITIONED RECALIBRATION MECHANISM

To ensure geometric–semantic consistency, UniSplat introduces a **pose-conditioned recalibration mechanism**. This component aligns predictions from different heads by reprojecting 3D outputs into the image plane using estimated camera parameters and minimizing their discrepancy with 2D renderings, thereby enforcing cross-task coherence across modalities.

UniSplat produces two complementary types of 3D predictions: explicit Gaussian fields ($\mathcal{G}_{\text{app}}$ for appearance and $\mathcal{G}_{\text{sem}}$ for semantics) and per-view 3D point maps $\mathbf{P}_v$. To ensure consistency among these predictions, the recalibration mechanism projects 3D maps back into 2D and compares them with fields rendered from Gaussians. This procedure forces all components to converge toward a unified and consistent scene representation.

**Geometric Recalibration.** For each view $v$, the 3D point map $\mathbf{P}_v$ is projected to 2D using the refined camera parameters $C_{\text{final}}^v$. We then measure consistency with the rendered RGB image from the appearance field $\mathcal{G}$app via a reprojection loss:

$$\mathcal{L}_{\text{recalib\_geo}} = \sum_{v=1}^{V} \sum_{j=1}^{H \times W} \|\mathbf{q}_j^v - \pi(C_{\text{final}}^v, \mathbf{p}_j^v)\|, \tag{9}$$

where $\mathbf{p}_j^v \in \mathbf{P}_v$ is the 3D point at pixel $j$, $\mathbf{q}_j^v$ is its 2D projection, and $\pi(\cdot)$ the projection operator.

**Semantic Recalibration.** To align semantic information, we build a 3D semantic point map $\mathbf{P}_v^{\text{sem}}$ by associating each 3D point in $\mathbf{P}_v$ with its semantic label. Projecting $\mathbf{P}_v^{\text{sem}}$ to 2D yields a semantic projection $F_v^{\text{proj}}$, which is aligned with the rendered semantic map $F_v^{\text{rend}}$ from $\mathcal{G}_{\text{sem}}$ using cosine similarity:

$$\mathcal{L}_{\text{recalib\_sem}} = \sum_{v=1}^{V} \sum_{j=1}^{H \times W} \left(1 - \frac{F_v^{\text{proj}}(j) \cdot F_v^{\text{rend}}(j)}{\|F_v^{\text{proj}}(j)\| \cdot \|F_v^{\text{rend}}(j)\|}\right). \tag{10}$$

The overall recalibration objective combines both terms:

$$\mathcal{L}_{\text{recalib}} = \lambda_{\text{recalib\_geo}} \mathcal{L}_{\text{recalib\_geo}} + \lambda_{\text{recalib\_sem}} \mathcal{L}_{\text{recalib\_sem}}. \tag{11}$$

with $\lambda_{\text{recalib\_geo}}$ and $\lambda_{\text{recalib\_sem}}$ controlling their relative contributions. By enforcing both geometric and semantic alignment in the 2D image plane, the recalibration mechanism ensures consistency between geometry and semantics, which is crucial for producing coherent 3D scene representations

## 3.5 TRAINING OBJECTIVES

Self-supervision from input views provides an essential learning signal, but it alone is insufficient for reliable 3D modeling. Thus, we adopt a composite objective that combines self-supervision with knowledge distillation, leveraging geometric and semantic priors from large-scale pre-trained foundation models to strengthen learning while avoiding the need for expensive 3D labels. The overall objective is a weighted sum of four terms:

$$\mathcal{L}_{\text{total}} = \lambda_{\text{rgb}} \mathcal{L}_{\text{rgb}} + \lambda_{\text{sem}} \mathcal{L}_{\text{sem}} + \lambda_{\text{geo}} \mathcal{L}_{\text{geo}} + \lambda_{\text{recalib}} \mathcal{L}_{\text{recalib}}, \tag{12}$$

where the $\lambda_*$ terms balance the contributions of each component.

**Photometric Reconstruction Loss** ensures that the rendered appearance $\hat{I}_v$ from $\mathcal{G}_{\text{app}}$ matches the input views $I_v$. It combines an L1 loss with the LPIPS perceptual metric for image quality:

$$\mathcal{L}_{\text{rgb}} = \sum_{v=1}^{V} \left(\|\hat{I}_v - I_v\|_2 + \lambda_{\text{LPIPS}} \cdot \text{LPIPS}(\hat{I}_v, I_v)\right). \tag{13}$$

**Semantic Distillation Loss** transfers open-vocabulary semantic knowledge from a frozen 2D vision–language model (VLM) into the 3D semantic Gaussians. For each view $v$, we extract a semantic feature map $F_v^{\text{VLM}}$ from the input image $I_v$ using the VLM's image encoder (*e.g.*, LSeg) and align it with the rendered semantic feature map $F_v^{\text{render}}$. The loss is defined as one minus the cosine similarity between these features:

$$\mathcal{L}_{\text{sem}} = \sum_{v=1}^{V} \sum_{j=1}^{H \times W} \left(1 - \frac{F_v^{\text{render}}(j) \cdot F_v^{\text{VLM}}(j)}{\|F_v^{\text{render}}(j)\| \cdot \|F_v^{\text{VLM}}(j)\|}\right). \tag{14}$$

**Geometric Prior Loss.** Following (Jiang et al., 2025b), we transfer geometric knowledge from a frozen VGGT teacher, which provides pseudo ground-truth camera parameters $\tilde{c}_i$ and point maps $\tilde{P}_j^v$, to regularize camera estimation and strengthen 3D structure learning. Camera parameters are regularized via a Huber loss:

$$\mathcal{L}_{\text{pose}} = \sum_{i=1}^{V} \|\tilde{c}_i - c_i\|_\epsilon, \tag{15}$$

Scene geometry is distilled using:

$$\mathcal{L}_{\text{point}} = \sum_{v=1}^{V} \sum_{j=1}^{H \times W} \widetilde{\text{Conf}}_j^v \cdot \|\tilde{P}_j^v - P_j^v\| + \|\widetilde{\text{Conf}}_j^v - \text{Conf}_j^v\|, \tag{16}$$

where $\widetilde{\text{Conf}}_j^v$ denotes the confidence score. The full geometric prior loss combines these terms:

$$\mathcal{L}_{\text{geo}} = \lambda_{\text{pose}} \mathcal{L}_{\text{pose}} + \lambda_{\text{point}} \mathcal{L}_{\text{point}}. \tag{17}$$

where $\lambda_{\text{pose}}$ and $\lambda_{\text{point}}$ are hyperparameters.

Table 1: **Quantitative Comparison on ScanNet.** We evaluate performance on novel view synthesis, depth estimation, and open-vocabulary semantic segmentation.

| Method | Recon. Time↓ | | Source View | | | | Target View | | | | |
| | SfM | Per-Scene | mIoU↑ | Acc.↑ | rel↓ | $\tau$ ↑ | mIoU↑ | Acc.↑ | PSNR↑ | SSIM↑ | LPIPS↓ |
|---|---|---|---|---|---|---|---|---|---|---|---|
| LSeg | N/A | N/A | 0.4701 | 0.7891 | - | - | 0.4819 | 0.7927 | - | - | - |
| NeRF-DFF | 20.52s | 1min | 0.4540 | 0.7173 | 27.68 | 9.61 | 0.4037 | 0.6755 | 19.86 | 0.6650 | 0.3629 |
| Feature-3DGS | 20.52s | 18mins | 0.4453 | 0.7276 | 12.95 | 21.07 | 0.4223 | 0.7174 | 24.49 | 0.8132 | 0.2293 |
| PixelSplat | | 0.064s | - | - | - | - | - | - | 24.89 | 0.8392 | 0.1641 |
| LSM | | 0.108s | 0.5034 | 0.7740 | 3.38 | 67.77 | 0.5078 | 0.7686 | 24.39 | 0.8072 | 0.2506 |
| Ours | | 0.041s | **0.5563** | **0.8277** | **3.10** | **69.13** | **0.5625** | **0.8334** | **25.65** | **0.8782** | **0.1353** |

## 4 EXPERIMENTS

### 4.1 EXPERIMENTAL SETUP

To comprehensively evaluate the effectiveness of our unified 3D representation, we test UniSplat in two distinct domains: (1) **traditional 3D vision tasks** to assess the quality of scene reconstruction, and (2) **embodied AI tasks** to evaluate the utility of the learned features as a visual backbone for downstream robotic control policies.

**3D Vision Tasks.** Following LSM, we evaluate UniSplat's scene understanding capabilities on 40 unseen scenes from the ScanNet (Dai et al., 2017) dataset. We assess three core tasks: Novel View Synthesis (PSNR, SSIM, LPIPS), Open-Vocabulary 3D Segmentation (mIoU, mAcc), and Depth Estimation (Abs Rel, Inlier Ratio). Furthermore, to evaluate rendering quality, we train our model on the RealEstate10K (Zhou et al., 2018) datasets. Figure 3 shows a qualitative comparison of the novel view synthesis on RealEstate10K. More details and results are provided in the Appendix.

**Embodied AI Tasks.** We use the pre-trained ViT encoder from UniSplat as a frozen feature extractor and evaluate it on the largest-scale embodied intelligence benchmark, which spans 268 tasks across 8 simulators. The evaluation covers both **single-task** (VC-1 (Majumdar et al., 2023), Franka Kitchen (Gupta et al., 2019), Meta-World (Yu et al., 2020)) and **language-conditioned multi-task** (RLBench (James et al., 2020), LIBERO (Liu et al., 2023)) scenarios, utilizing a variety of policies including MLPs, Diffusion, and Transformers. We compare against a diverse set of leading visual representation learning models, including **vision-centric** (MAE (He et al., 2022), DINOv2 (Oquab et al., 2023)), **multi-modal** (CLIP (Radford et al., 2021), EVA (Fang et al., 2023), InternViT (Chen et al., 2024b)), and **embodied-specific** (VC-1, MVP (Radosavovic et al., 2023), SPA) approaches.

**Implementation Details.** UniSplat is built upon a ViT-L backbone pre-trained on ScanNet and ScanNet++ Yeshwanth et al. (2023), equipped with a multi-task decoder. To circumvent explicit 3D supervision, we leverage LSeg and VGGT teachers to generate pseudo ground-truth semantics and geometry. All experiments are optimized using AdamW with a base learning rate of $1 \times 10^{-4}$ and a 30-epoch warm-up schedule. Training is conducted for 300 epochs on 4 NVIDIA A100 GPUs. For fair comparison with baseline methods, the input resolution is fixed at $256 \times 256$. The dual-masking strategy adopts masking ratios of 0.5 for both encoder and decoder. Each anchor Gaussian yields 10 derived Gaussians, and the number of coarse Gaussian tokens $T_{\text{coarse}}$ is set to 256. The loss weights are configured as follows: $\lambda_{\text{rgb}} = 1.0$, $\lambda_{\text{LPIPS}} = 0.05$, $\lambda_{\text{sem}} = 0.3$, $\lambda_{\text{geo}} = 1.2$, $\lambda_{pose} = 10.0$, $\lambda_{\text{point}} = 1.5$, $\lambda_{\text{recalib\_geom}} = 0.001$, $\lambda_{\text{recalib\_geo}} = 0.5$, and $\lambda_{\text{recalib}} = 1.0$.

### 4.2 RESULTS ON 3D VISION TASKS

Table 1 summarizes results against pose-based baselines requiring SfM and/or per-scene optimization (NeRF-DFF (Kobayashi et al., 2022), Feature-3DGS, PixelSplat) and the strong pose-free baseline LSM.

**Open-Vocabulary 3D Segmentation.** As presented in Table 1 and Figure 2, UniSplat sets a new pose-free state of the art. On source views, it reaches 0.5563 mIoU and 0.8277 mAcc, surpassing LSM by +5.3/+5.4 points. On target views, it attains 0.5625 mIoU and 0.8334 mAcc, improving over LSM by +5.5/+6.5 points and exceeding the 2D LSeg baseline while providing cross-view consistency that 2D methods lack.

**Novel View Synthesis.** Without SfM or per-scene fitting, UniSplat delivers the best image quality among compared methods: 25.65 PSNR, 0.8782 SSIM, and 0.1353 LPIPS on target views. This outperforms LSM, the generalizable PixelSplat that assumes known cameras, and the opti-

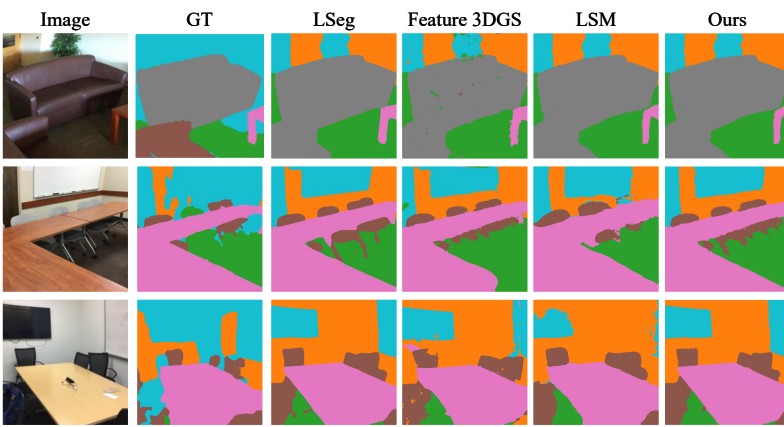

|  | Image | GT | LSeg | Feature 3DGS | LSM | Ours |
|--|-------|-----|------|--------------|-----|------|

Figure 2: Qualitative comparison of novel-view segmentation on ScanNet.

mized Feature-3DGS. Gains indicate that geometry-aware masking and progressive semantic-to-appearance rendering yield sharper, more faithful novel views.

**Depth Estimation.** UniSplat achieves a 3.10 Absolute Rel and 69.13% inlier ratio $\tau$ on source views, improving over LSM. Methods optimized per scene remain far behind in geometry.

**Efficiency and Prerequisites.** UniSplat is feed-forward and pose-free (no SfM, no per-scene optimization), while NeRF-DFF and Feature-3DGS require both and PixelSplat assumes camera poses. Relative to LSM, UniSplat consistently improves segmentation, NVS, and depth while preserving the same deployment simplicity. These results validate that unifying semantics, geometry, and appearance with camera-centric recalibration improves both 3D understanding and rendering quality.

Table 2: **Comparison of different representation learning methods.** The number in parentheses denotes the number of tasks.

| Benchmark | Method | Vision-Centric | | Multi-Modal | | | Embodied-Specific | | | Ours |
|-----------|--------|----------------|--------|-------------|-----|----------|-------------------|------|------|------|
|  |  | MAE | DINOV2 | CLIP | EVA | InternViT | MVP | VC-1 | SPA | Ours |
| VC-1 | AD (2) | 58.0±2.0 | 47.3±3.1 | 48.7±3.1 | 58.0±6.0 | 53.3±3.1 | 53.3±4.2 | 54.0±4.0 | 60.0±4.0 | **61.7±4.3** |
|  | MW (5) | 90.0±4.6 | 84.0±3.7 | 77.1±3.2 | 90.7±0.9 | 84.0±3.7 | 93.6±5.2 | 87.5±3.8 | 93.3±2.0 | **94.3±3.1** |
|  | DMC (5) | 74.4±1.8 | 64.5±2.5 | 53.9±3.6 | 62.7±2.8 | 53.3±0.4 | 69.4±2.6 | 65.3±3.6 | 71.1±5.0 | **75.8±4.5** |
|  | TF (2) | 73.0±0.5 | 68.5±0.4 | 56.1±1.6 | 67.2±0.2 | 65.2±1.6 | 73.2±0.8 | 70.9±1.1 | 73.6±2.0 | **75.6±1.7** |
| RLBench | Group 1 (35) | 78.3 | 78.2 | 76.8 | 75.2 | 74.1 | 76.2 | 80.1 | 80.5 | **81.2** |
|  | Group 2 (36) | 57.7 | 56.1 | 55.7 | 57.0 | 54.9 | 56.3 | 55.7 | 61.2 | **63.3** |
| Meta-World (48) | | 67.8±1.7 | 56.3±0.6 | 66.7±1.7 | 63.7±1.3 | 57.5±1.7 | 66.4±1.7 | 68.6±1.5 | 69.2±1.7 | **70.9±1.** |
| LIBERO | Object (10) | 71.7±13.1 | 64.7±9.9 | 50.2±7.0 | 73.2±6.0 | 67.7±6.0 | 63.7±4.8 | 69.7±7.2 | 76.7±5.3 | **78.4±6.1** |
|  | Spatial (10) | 57.2±2.9 | 36.3±11.8 | 32.2±0.6 | 59.3±7.7 | 48.3±6.4 | 58.0±6.2 | 50.5±7.5 | 50.0±3.8 | **59.7±5.8** |
|  | Goal (10) | 54.3±6.0 | 22.2±2.3 | 30.3±3.2 | 56.8±2.9 | 58.8±4.5 | 63.8±2.8 | 57.5±6.6 | 65.3±2.5 | **67.3±2.3** |
|  | 10 (10) | 41.2±4.5 | 28.3±3.0 | 27.5±3.9 | 43.3±2.8 | 38.2±1.3 | 39.0±0.9 | 39.7±3.5 | 40.2±3.6 | **42.4±3.5** |
|  | 90 (90) | 29.9±2.0 | 27.5±2.2 | 29.4±2.0 | 31.3±2.3 | 23.8±1.8 | 32.1±3.5 | 30.6±3.3 | 32.2±1.6 | **34.7±2.7** |
| Franka-Kitchen (5) | | 42.7±2.6 | 40.9±6.4 | 30.8±3.3 | 37.3±1.3 | 28.5±1.7 | 34.3±6.1 | 37.5±3.5 | 40.6±1.9 | **44.5±2.6** |

### 4.3 RESULTS ON EMBODIED AI TASKS

As shown in Table 2, UniSplat consistently outperforms vision-centric (MAE, DINOv2), multi-modal (CLIP, EVA, InternViT), and embodied-specific (MVP, VC-1, SPA) baselines. On VC-1, UniSplat attains top scores across AD, MW, DMC, and TF. It sets new records on RLBench (81.2%/63.3% for Group 1/2) and Meta-World (70.9%), while achieving strong gains in all LIBERO splits, including 78.4% on Object, 59.7% on Spatial, and 67.3% on Goal, with robust results on LIBERO-90. On Franka Kitchen, it reaches 44.5%, surpassing prior methods. These results show that UniSplat's unified 3D representation transfers effectively to varied visuomotor control tasks without task-specific tuning.

### 4.4 ABLATION STUDIES

**Ablation on Key Component.** Table 3 shows every component matters. Removing semantic distillation collapses segmentation while keeping appearance nearly intact, underscoring its role for open-vocabulary semantics. Camera recalibration and geometric prior losses are critical for geometry and rendering. Geometry-aware dual mask and coarse-to-fine Gaussians splatting strategy yield consistent gains. Disabling self-supervised learning degrades all metrics.

Table 3: **Ablation on Core Components.** Each row removes one component to evaluate its contribution.

| Variant | mIoU↑ | Acc.↑ | PSNR↑ | SSIM↑ |
|---|---|---|---|---|
| Full UniSplat | **0.5625** | **0.8334** | **25.65** | **0.8782** |
| w/o Self-sup. | 0.5263 | 0.8110 | 24.40 | 0.8092 |
| w/o Dual Mask | 0.5462 | 0.8275 | 24.74 | 0.8373 |
| w/o Coarse-to-Fine | 0.5374 | 0.8239 | 24.93 | 0.8452 |
| w/o $\mathcal{L}_{recalib}$ | 0.5117 | 0.8086 | 24.35 | 0.8067 |

Table 4: **Ablation on Data Scale.** We evaluate the effect of progressively adding more training data from different datasets.

| Exp ID | Datasets | mIoU↑ | Acc.↑ | PSNR↑ | SSIM↑ |
|---|---|---|---|---|---|
| (1) | ScanNet | 0.5603 | 0.8297 | 25.48 | 0.8724 |
| (2) | (1) + ScanNet++ | 0.5625 | 0.8334 | 25.65 | 0.8782 |
| (3) | (2) + RealEstate10K | 0.5717 | 0.8414 | 25.79 | 0.8837 |
| (4) | (3) + DL3DV | **0.5755** | **0.8437** | **25.83** | **0.8916** |

Table 5: **Ablation on the Number of Input Views.** We evaluate the effect of varying the number of input views on reconstruction and segmentation performance.

| Number of Views | mIoU↑ | Acc.↑ | PSNR↑ | SSIM↑ | LPIPS↓ |
|---|---|---|---|---|---|
| 3 | 0.5574 | 0.8292 | 25.83 | 0.8797 | 0.1303 |
| 6 | 0.5846 | 0.8451 | 26.75 | 0.8827 | 0.1234 |
| 8 | 0.6027 | 0.8454 | 27.03 | 0.8855 | 0.1126 |
| 10 | 0.6227 | 0.8514 | 27.12 | 0.8862 | 0.1156 |

**Scale Up with More Training Data.** Table 4 demonstrates that expanding the training dataset can significantly enhance model generalization and robustness. Larger, diverse datasets expose the model to varied patterns, reducing overfitting and improving performance on unseen data.

**Ablation on the Number of Input View.** As shown in Table 5, increasing input views improves both segmentation and reconstruction. More views provide richer geometric and appearance cues, enhancing scene understanding and rendering quality. Gains diminish beyond 8 views, indicating that while additional viewpoints help, the marginal benefit reduces once sufficient coverage is achieved.

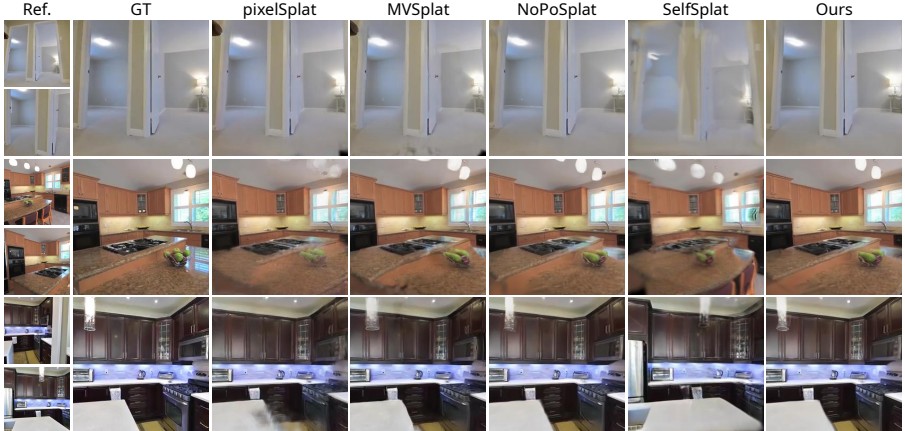

Figure 3: Qualitative comparison of novel view synthesis on RealEstate10k.

## 5 CONCLUSION

We introduced UniSplat, a self-supervised framework that learns unified 3D representations directly from unposed multi-view images. UniSplat addresses the key limitations of prior methods through three complementary components: a dual-masking strategy that strengthens geometry induction by enforcing structure reasoning from incomplete cues, a coarse-to-fine Gaussian splatting strategy that progressively refines scene appearance to capture both global structure and fine detail, and a pose-conditioned recalibration mechanism that enforces geometric–semantic consistency by aligning multi-head predictions in a shared spatial frame. Together, these designs enable UniSplat to produce coherent, high-fidelity 3D representations that are robust to sparse-view settings and transferable across domains. Extensive experiments on both 3D vision benchmarks and embodied AI tasks confirm its effectiveness and versatility. Future work will explore scaling to larger, more diverse datasets and integrating language-scene interaction for richer embodied intelligence.

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

## A    THE USE OF LARGE LANGUAGE MODELS

In this paper, Large Language Models were used solely for minor language polishing and grammar correction. No LLMs contributed to research ideation, experimental design, or substantive content generation.

## B    MORE IMPLEMENTATION DETAILS

### B.1    3D VISION TASKS IMPLEMENTATION DETAILS

We adopt two complementary experimental settings for evaluating 3D vision tasks, each emphasizing different aspects of the unified 3D representation:

**LSM-based Setting (Semantic–Geometry–Appearance Unification).** Following the Large Spatial Model (LSM)(Fan et al., 2024) protocol, we train and evaluate on indoor scene datasets (ScanNet and ScanNet++), where the model must jointly predict geometry, appearance, and semantics from *unposed* multi-view images. This setting stresses *semantic consistency* across views and the ability to lift 2D features into a coherent 3D semantic field.

**NoPoSplat-based Setting (Pose-free High-fidelity Reconstruction).** Following NoPoSplat (Ye et al., 2025), we train and evaluate on large-scale video datasets (RealEstate10K and ACID) for *pose-free* novel view synthesis and relative pose estimation. This setting emphasizes *geometric fidelity* and robustness to sparse, wide-baseline inputs without any pose supervision. The corresponding results are shown in C.1.

### B.2    EMBODIED TASKS IMPLEMENTATION DETAILS

To evaluate the effectiveness of UniSplat as a visual backbone for embodied AI, we follow the large-scale embodied evaluation protocol introduced in (Zhu et al., 2025), which spans **268 tasks** across **8 simulators** and covers both single-task and language-conditioned multi-task scenarios. In all experiments, the UniSplat encoder is **frozen** and only the downstream policy network is trained, ensuring a fair comparison of representation quality.

**Single-task Benchmarks.** We include three representative single-task settings:

- **VC-1** (Majumdar et al., 2023): 14 tasks from four simulators — Adroit (AD) (Kumar, 2016), Meta-World (MW) (Yu et al., 2020), DMControl (DMC) (Tunyasuvunakool et al., 2020), and TriFinger (TF) (Wüthrich et al., 2020). Policies are 3-layer MLPs trained with 100 demonstrations per task (25 for MW) and evaluated over 50 rollouts using fixed seeds (100, 200, 300). The `[CLS]` token from UniSplat serves as the observation feature.
- **Franka Kitchen** (Gupta et al., 2019): 5 manipulation tasks in a MuJoCo kitchen scene, each with two camera viewpoints and three seeds. Policies are 2-layer MLPs trained on 25 demonstrations per task.
- **Meta-World**: 48 diverse manipulation tasks. We adopt the Diffusion Policy (Chi et al., 2023) following (Ze et al., 2024), training with 10 demonstrations and evaluating over 20 rollouts per task.

**Language-conditioned Multi-task Benchmarks.** We also evaluate on two multi-task suites with natural language instructions:

- **RLBench** (James et al., 2020): 71 executable tasks split into two groups according to Polar-Net (Chen et al., 2023) categories (35 and 36 tasks). We use RVT-2 (Goyal et al., 2024) as the policy backbone, replacing its CNN encoder with our frozen UniSplat encoder. Each task has 100 demonstrations for training and 25 rollouts for evaluation.
- **LIBERO** (Liu et al., 2023): 130 tasks across five suites (Spatial, Object, Goal, LIBERO-10, LIBERO-90). We train the official transformer-based language-conditioned policy with 20 demonstrations per task, no data augmentation, and pre-extracted visual features from UniSplat.

**Policy Training and Evaluation.** For all settings, we adhere to the training hyperparameters and evaluation protocols of the respective benchmarks to ensure comparability with prior work (Zhu

et al., 2025). The frozen UniSplat encoder outputs either the `[CLS]` token (for MLP/Diffusion policies) or unpatchified feature maps (for transformer-based policies)..

## C   MORE EXPERIMENTAL ANALYSIS

### C.1   MORE RESULTS ON 3D VISION TASKS

**Novel View Synthesis.** As shown in Table 6, we compare our method with several state-of-the-art methods on the RealEstate10K dataset. Our method outperforms all previous pose-free methods and even surpasses some pose-required methods, demonstrating the effectiveness of our unified 3D representation and training strategy. Qualitative results are shown in Figure 4,Figure 5 and Figure 6.

Table 6: Performance comparison of novel view synthesis on the RE10K dataset.

| Method | Small | | | Medium | | | Large | | | Average | | |
|---|---|---|---|---|---|---|---|---|---|---|---|---|
| | PSNR ↑ | SSIM ↑ | LPIPS ↓ | PSNR ↑ | SSIM ↑ | LPIPS ↓ | PSNR ↑ | SSIM ↑ | LPIPS ↓ | PSNR ↑ | SSIM ↑ | LPIPS ↓ |
| *Pose-Required* | | | | | | | | | | | | |
| pixelSplat | 20.277 | 0.719 | 0.265 | 23.726 | 0.811 | 0.180 | 27.152 | 0.880 | 0.121 | 23.859 | 0.808 | 0.184 |
| MVSplat | 20.371 | 0.725 | 0.250 | 23.808 | 0.814 | 0.172 | 27.466 | 0.885 | 0.115 | 24.012 | 0.812 | 0.175 |
| *Supervised Pose-Free* | | | | | | | | | | | | |
| MASt3R | 16.305 | 0.516 | 0.451 | 18.106 | 0.561 | 0.377 | 17.975 | 0.524 | 0.402 | 17.617 | 0.539 | 0.403 |
| CoPoNeRF | 17.393 | 0.585 | 0.462 | 18.813 | 0.616 | 0.392 | 20.464 | 0.652 | 0.318 | 18.938 | 0.619 | 0.388 |
| Splatt3R | 17.789 | 0.582 | 0.375 | 18.828 | 0.607 | 0.330 | 19.243 | 0.593 | 0.317 | 18.688 | 0.593 | 0.317 |
| NoPoSplat | 22.514 | 0.784 | 0.210 | 24.899 | 0.839 | 0.160 | 27.411 | 0.883 | 0.119 | 25.033 | 0.838 | 0.160 |
| *Self-Supervised Pose-Free* | | | | | | | | | | | | |
| SelfSplat | 14.828 | 0.543 | 0.469 | 18.857 | 0.679 | 0.328 | 23.338 | 0.798 | 0.208 | 19.152 | 0.680 | 0.328 |
| **Ours** | **22.765** | **0.789** | **0.205** | **25.246** | **0.845** | **0.156** | **27.872** | **0.891** | **0.113** | **25.397** | **0.843** | **0.157** |

**Relative Pose Estimation.** We evaluate the relative pose estimation performance of our method on the RealEstate10K and ACID datasets, following the protocol in (Ye et al., 2025). As shown in Table 7, our method outperforms all baselines across all thresholds, demonstrating the effectiveness of our unified 3D representation in capturing accurate camera poses from unposed multi-view images.

Table 7: Pose estimation performance in AUC with various thresholds on RE10K and ACID datasets.

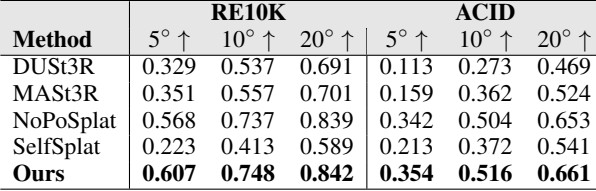

| Method | RE10K | | | ACID | | |
|---|---|---|---|---|---|---|
| | 5° ↑ | 10° ↑ | 20° ↑ | 5° ↑ | 10° ↑ | 20° ↑ |
| DUSt3R | 0.329 | 0.537 | 0.691 | 0.113 | 0.273 | 0.469 |
| MASt3R | 0.351 | 0.557 | 0.701 | 0.159 | 0.362 | 0.524 |
| NoPoSplat | 0.568 | 0.737 | 0.839 | 0.342 | 0.504 | 0.653 |
| SelfSplat | 0.223 | 0.413 | 0.589 | 0.213 | 0.372 | 0.541 |
| **Ours** | **0.607** | **0.748** | **0.842** | **0.354** | **0.516** | **0.661** |

### C.2   MORE ABLATION STUDIES

**Ablation on Mask Strategy.** As shown in Table 8, Croco masking (Weinzaepfel et al., 2022) surpasses random masking under similar settings but remains weaker than our geometry-aware two-stage masking strategy. For random masking, overly low $\rho_e$ yields insufficient spatial reasoning, while excessively high $\rho_e$ hinders learning. The geometry-aware mask consistently improves both segmentation and reconstruction, indicating balanced masking facilitates optimal representation learning.

**Ablation on the Number of Gaussian Latent Tokens.** Table 9 shows that increasing the number of Gaussian latent tokens from 64 to 128 improves mIoU, accuracy, and perceptual quality. 256 tokens give the best overall metrics with marginal gains over 128, while 512 tokens slightly degrade performance, suggesting diminishing returns and possible overfitting. An intermediate token count balances reconstruction fidelity and segmentation accuracy.

### C.3   DOWNSTREAM TASKS

We further validate the utility of UniSplat as a general 3D visual backbone on EmbodiedScan (Wang et al., 2024c), a large-scale, ego-centric multi-modal 3D perception benchmark with oriented 3D boxes, semantic occupancy, and language prompts. We follow the official data organization, view

Table 8: **Ablation on Mask Strategy.** We evaluate the effect of varying mask strategies on reconstruction and segmentation performance.

| Type of $\mathcal{M}_e$ | $\rho_e$ | Type of $\mathcal{M}_d$ | $\rho_d$ | mIoU↑ | Acc.↑ | PSNR↑ | SSIM↑ |
|---|---|---|---|---|---|---|---|
| Random | 0.50 | N/A | 0 | 0.5502 | 0.8275 | 24.84 | 0.8373 |
| Croco | 0.90 | N/A | 0 | 0.5531 | 0.8385 | 25.12 | 0.8510 |
| Random | 0.50 | 3D GS | 0.30 | 0.5511 | 0.8280 | 25.42 | 0.8721 |
| Random | 0.50 | 3D GS | 0.50 | **0.5625** | **0.8334** | **25.65** | **0.8782** |
| Random | 0.50 | 3D GS | 0.75 | 0.5470 | 0.8265 | 25.10 | 0.8680 |
| Random | 0.75 | 3D GS | 0.50 | 0.5284 | 0.8120 | 24.55 | 0.8523 |
| Random | 0.75 | 3D GS | 0.75 | 0.5231 | 0.8065 | 24.21 | 0.8450 |

Table 9: **Ablation on the Number of Gaussian Latent Tokens.** We evaluate the effect of varying the number of Gaussian latent tokens on reconstruction and segmentation performance.

| Number of $T_{\text{coarse}}$ | mIoU↑ | Acc.↑ | PSNR↑ | SSIM↑ | LPIPS↓ |
|---|---|---|---|---|---|
| 64 | 0.5469 | 0.8237 | 25.05 | 0.8602 | 0.1478 |
| 128 | 0.5587 | 0.8284 | 25.33 | 0.8684 | 0.1413 |
| 256 | **0.5625** | **0.8334** | **25.65** | **0.8782** | **0.1353** |
| 512 | 0.5617 | 0.8312 | 25.53 | 0.8756 | 0.1371 |

sampling, and metric protocols to evaluate three downstream tasks: multi-view 3D detection, multi-view semantic occupancy prediction, and multi-view 3D visual grounding. To adapt UniSplat for specific tasks, we append lightweight task-specific heads to its pretrained multi-view transformer encoder. For 3D object detection, we attach a 3D detection head predicting oriented boxes (center, size, rotation) from UniSplat's fused multi-view geometric-semantic features. For semantic occupancy prediction, a 3D decoder is added, taking the voxelized dense features to predict semantic grids. For 3D visual grounding, we equip the 3D decoder with a cross-modal fusion transformer that integrates encoded language features with the 3D scene representation, followed by a grounding head sharing the detection architecture. This setup directly measures how well self-supervised 3D representations learned from unposed images generalize to complex indoor perception tasks without any depth supervision.

**Multi-view 3D Object Detection.** As shown in Table 10, UniSplat with RGB-only inputs consistently surpasses camera-only baselines and even strong RGB-D systems on EmbodiedScan, indicating that its unified geometric–semantic representation yields reliable oriented box estimates without depth supervision.

**Multi-view Semantic Occupancy Prediction.** As shown in Table 11, UniSplat delivers markedly better voxel-level semantics than prior RGB methods and is competitive with or exceeds RGB-D variants, reflecting dense, scene-consistent 3D priors learned from unposed images.

**Multi-view 3D Visual Grounding.** As shown in Table 12, UniSplat shows clear improvements over RGB-D baselines across overall, easy, and hard settings, demonstrating robust cross-modal alignment and spatial grounding from images alone.

Results show UniSplat provides consistent gains over camera-only baselines across all tasks, highlighting its effectiveness as a unified RGB-only 3D backbone.

Table 10: Multi-view 3D object detection results on EmbodiedScan.

| Method | Input | $AP_{25}$ | $AR_{25}$ | $AP_{50}$ | $AR_{50}$ |
|---|---|---|---|---|---|
| ImVoxelNet Rukhovich et al. (2022b) | RGB | 6.15 | 20.39 | 2.41 | 6.31 |
| VoteNet Qi et al. (2019) | Depth | 3.20 | 6.11 | 0.38 | 1.22 |
| FCAF3D Rukhovich et al. (2022a) | Depth | 9.07 | 44.23 | 4.11 | 20.22 |
| EmbodiedScan Wang et al. (2024c) | RGB-D | 16.85 | 51.07 | 9.77 | 28.21 |
| Ours | RGB | **28.69** | **62.24** | **15.34** | **39.57** |

## D  MORE VISUALIZATIONS

We present additional qualitative results to further illustrate the effectiveness of UniSplat across varying scene conditions and view overlaps, as shown in Figure 4–7.

Table 11: Multi-view semantic occupancy prediction results on EmbodiedScan.

| Method | Input | mIoU |
|---|---|---|
| OccNet (Tong et al., 2023) | RGB | 8.07 |
| SurroundOcc (Wei et al., 2023) | RGB | 9.10 |
| EmbodiedScan | RGB | 10.48 |
| EmbodiedScan | RGB-D | 19.97 |
| Ours | RGB | **27.45** |

Table 12: Multi-view 3D visual grounding results on EmbodiedScan.

| Method | Input | Overall | Easy | Hard |
|---|---|---|---|---|
| ScanRefer (Chen et al., 2020a) | RGB-D | 12.85 | 13.78 | 9.12 |
| BUTD-DETR (Jain et al., 2022) | RGB-D | 22.14 | 23.12 | 18.23 |
| L3Det (Zhu et al., 2023a) | RGB-D | 23.07 | 24.01 | 18.34 |
| EmbodiedScan | RGB-D | 25.72 | 27.11 | 20.12 |
| Ours | RGB | **36.88** | **38.13** | **31.42** |

# E DISCUSSION AND LIMITATIONS

**Limitations.** Although UniSplat achieves strong performance across diverse 3D vision and embodied AI tasks, several limitations remain. First, the framework still relies on pseudo-supervision from large pre-trained teacher models for geometry and semantics, which may propagate biases and inaccuracies from these teachers into the learned representation. Second, while our geometry-aware masking and coarse-to-fine splatting improve robustness to sparse unposed views, performance degrades in extremely limited or highly textureless scenes, indicating that geometry induction could be further strengthened. Third, our experiments are primarily conducted on indoor datasets; scaling to large-scale outdoor or highly dynamic environments may require additional adaptations, such as motion modeling or more robust pose estimation. Finally, although the hierarchical Gaussian representation improves efficiency compared to dense splatting, rendering and training remain computationally intensive relative to purely latent approaches, which may limit deployment in resource-constrained settings.

**Future Work.** Future research could address these limitations in several ways. One direction is to develop geometry and semantic priors that are learned in a fully self-supervised manner, reducing dependence on external teacher models. Another is to design adaptive masking and rendering strategies that adjust to scene complexity and viewpoint coverage, further improving robustness in sparse or degenerate cases. Extending UniSplat to handle dynamic, open-world environments and outdoor scenes would broaden its applicability, potentially requiring integration of temporal modeling and more generalizable camera estimation. Finally, incorporating large-scale language-scene interaction and multi-modal grounding could enable richer spatial reasoning and task understanding for embodied agents.

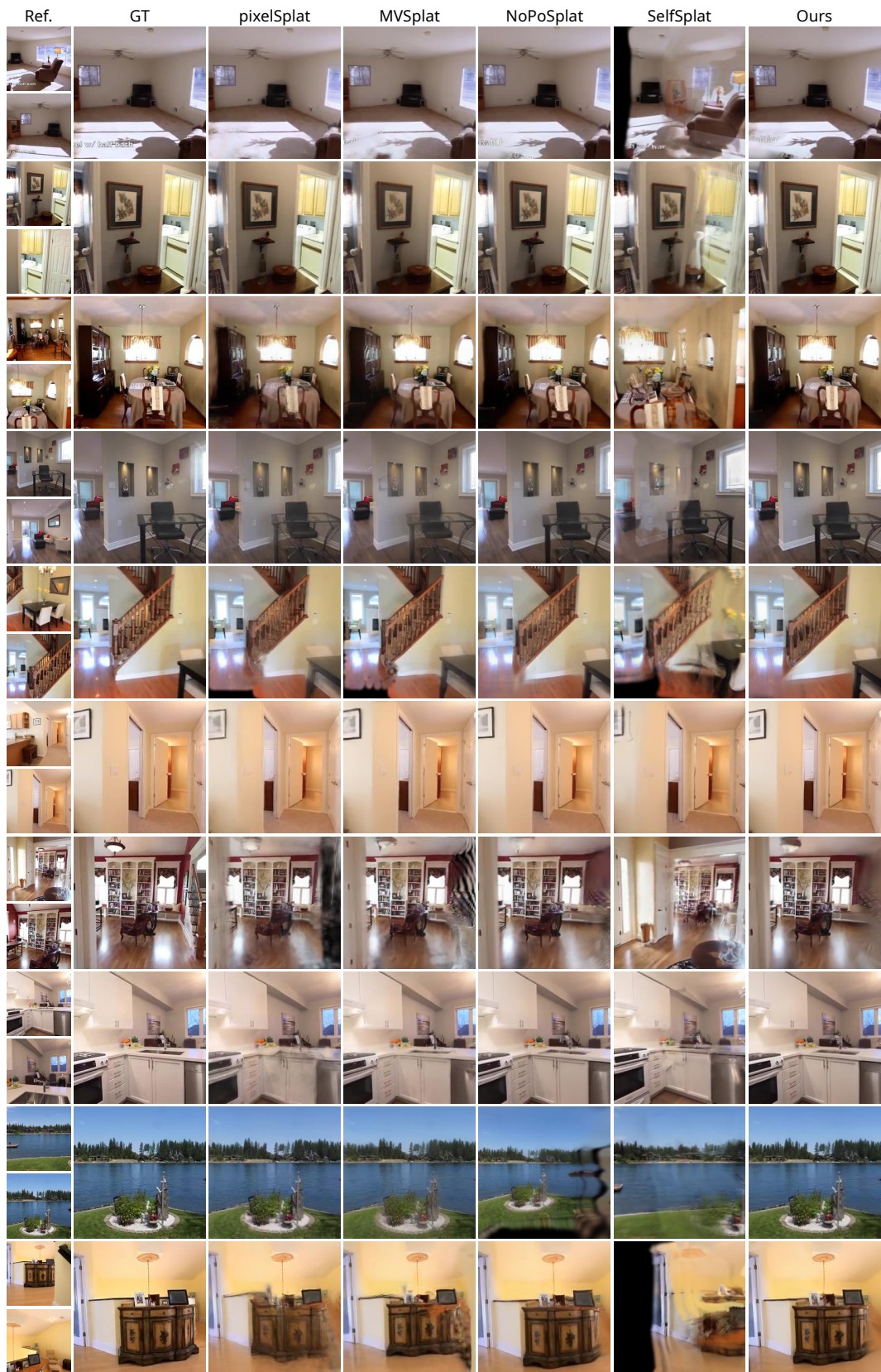

Figure 4: More qualitative comparisons on RE10K with small image overlap.

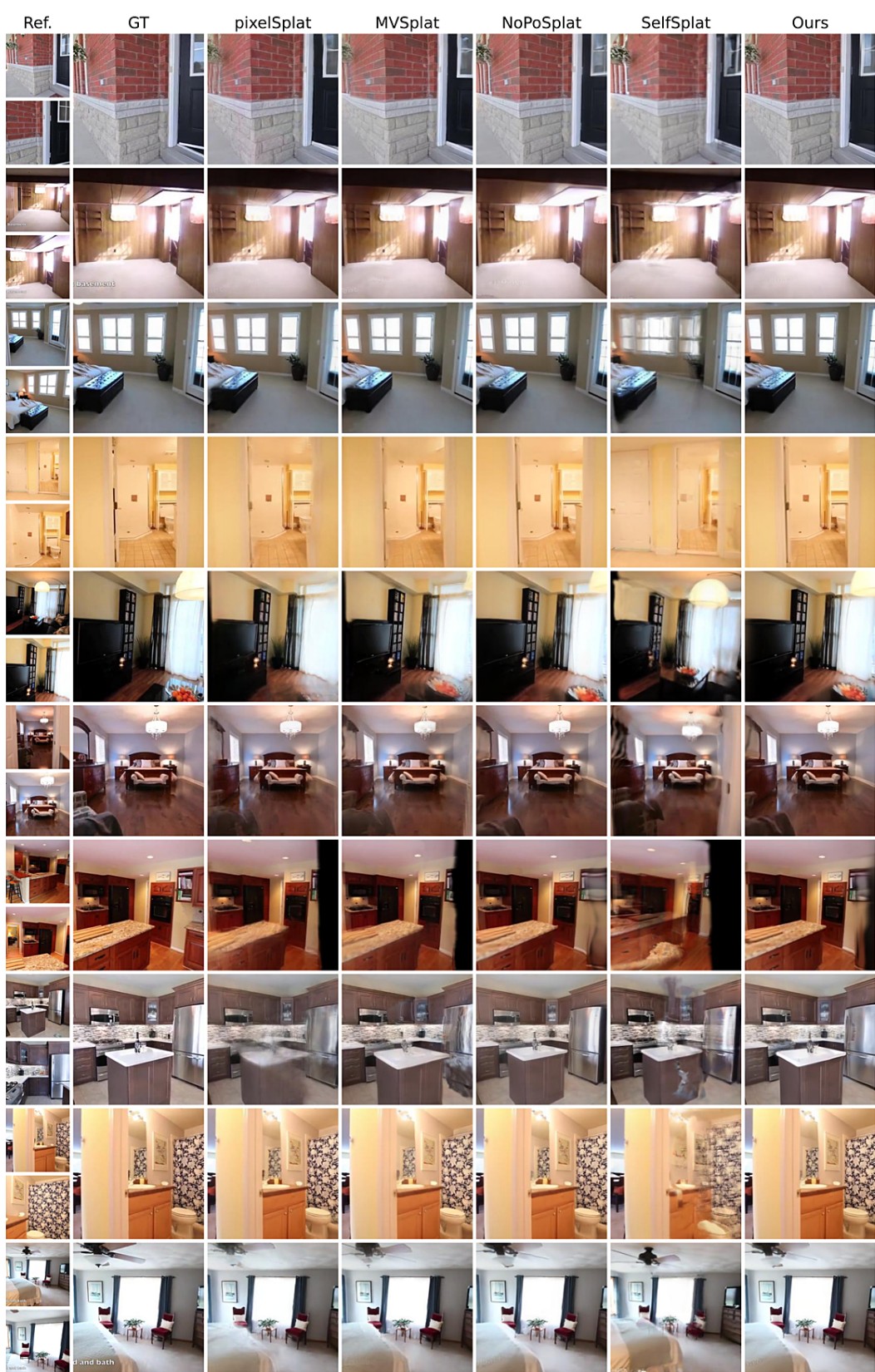

Figure 5: More qualitative comparisons on RE10K with medium image overlap.

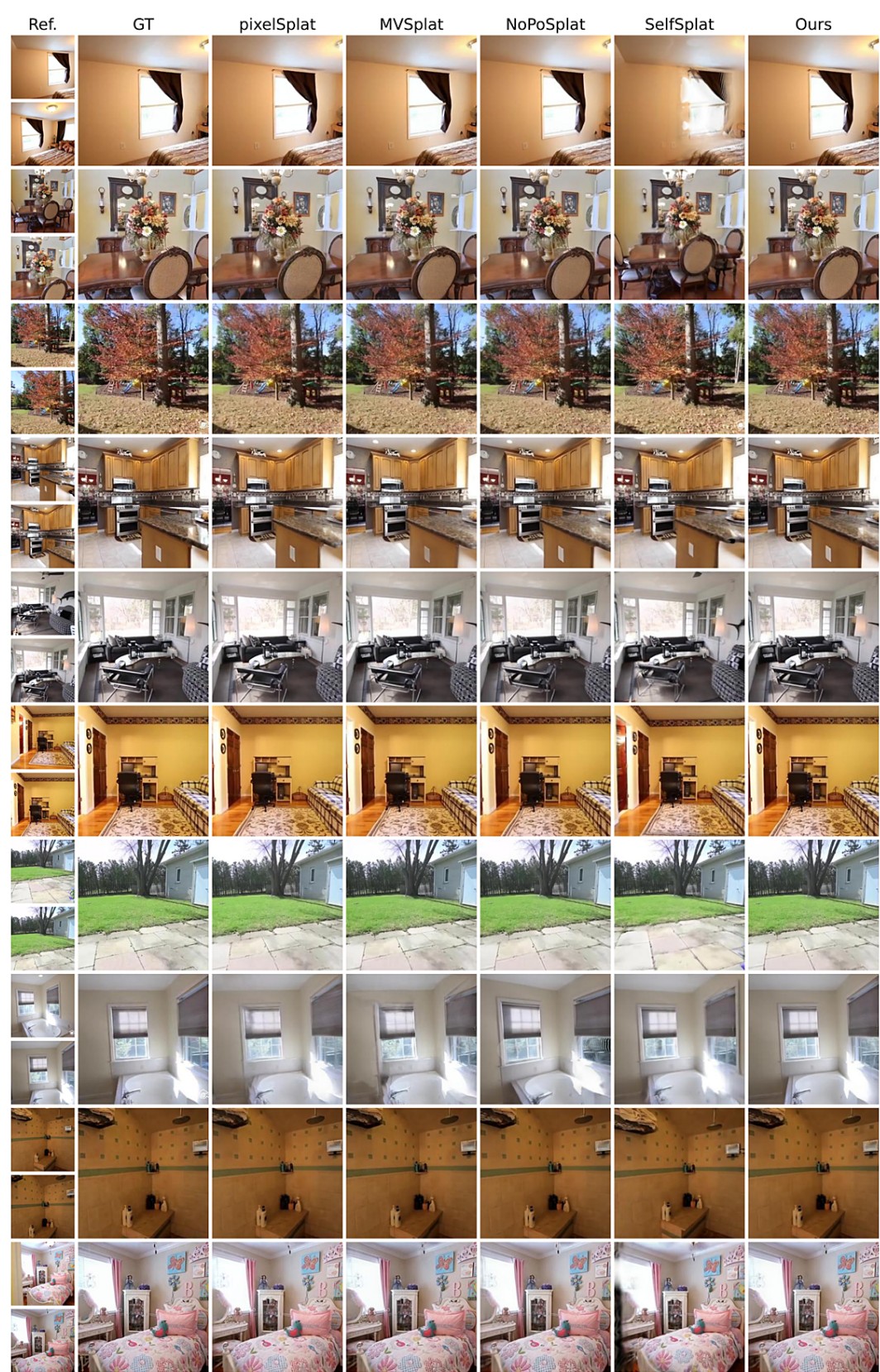

Figure 6: More qualitative comparisons on RE10K with large image overlap.

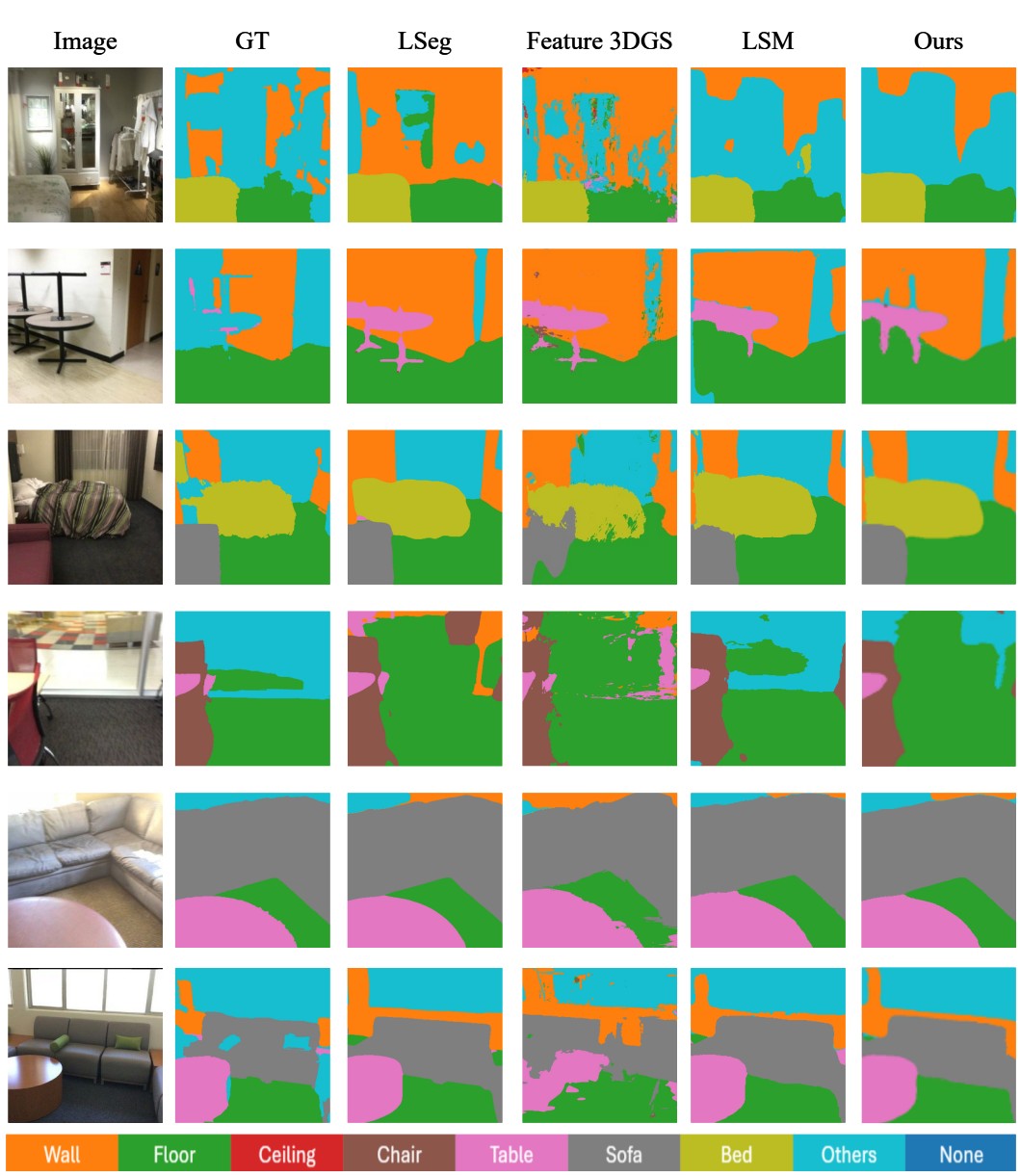

Figure 7: More qualitative comparison of novel-view segmentation on ScanNet.

