# OpenReview forum: "Toward Spatial Intelligence: A Unified Self-supervised Framework for 3D Representation Learning from Unposed Multi-View Images"
_ICLR.cc/2026/Conference — ICLR 2026 Conference Withdrawn Submission_

### Official Review · Reviewer_WiqJ · 2025-10-26

**Soundness:** 3
**Presentation:** 2
**Contribution:** 2
**Rating:** 4
**Confidence:** 3

**Summary:**

This paper introduces UniSplat, a self-supervised framework designed to learn a unified 3D representation of geometry, appearance, and semantics from unposed multi-view images. Specifically, the paper proposes:

(1) a *dual-masking strategy* that uses gaussian-guided geometric cues to extract robust geometric information from partial visual inputs;

(2) a *coarse-to-fine Gaussian splatting strategy* that progressively refines anchor, semantic, and appearance Gaussians, addressing the granularity mismatch between semantic and appearance representations in conventional 3D learning;

(3) a *pose-conditioned recalibration mechanism* that leverages VLM and VGGT to provide supervision for semantic and geometric alignment, improving their consistency.

Overall, the method shows excellent performance on both traditional 3D vision and embodied AI tasks, demonstrating its effectiveness and versatility.

**Strengths:**

1. The paper is well-written and logically structured, focusing on the unified self-supervised learning of geometry, appearance, and semantics in 3D representation learning.

2. By leveraging Gaussian splatting to provide geometric cues, the method effectively mitigates the issue of overfitting to trivial textures commonly seen in traditional approaches.

3. The large gap between semantic fields and appearance fields is progressively bridged through the introduction of anchors, enabling better alignment between the two.

4. The proposed method achieves strong performance across multiple benchmarks, with comprehensive evaluation metrics and thorough ablation studies, demonstrating the soundness and robustness of the approach.

**Weaknesses:**

1. Although geometry, appearance, and semantics are all important factors for unified 3D representation, considering all of them within a single framework may lead to a potential issue of being *broad but not deep*.

2. The experimental section lacks sufficient analysis or evidence to clearly demonstrate the advantages of UniSplat in achieving a unified representation across geometry, appearance, and semantic aspects.

3. The overall framework of UniSplat is relatively complex. Although it might be efficient in practice, it is unclear whether the method is easy to reproduce.

4. UniSplat involves a large number of hyperparameters, yet their selection process and sensitivity are not discussed or analyzed in the paper.

**Questions:**

1. Section 3.4 introduces the fusion strategy between geometry and semantics, while Section 3.3 describes the alignment between semantics and appearance. Beyond these, does UniSplat explicitly consider the direct relationship between geometry and appearance?
2. Although the paper claims that UniSplat is a self-supervised framework, its training relies on supervision signals from VLM and VGGT for semantic and geometric guidance, respectively. Does this contradict the self-supervised learning claim?
3. What is the source of the GT shown in Figure 2? In some cases (e.g., Case 2 and Case 3), the GT results appear even worse than other methods.
4. In Section 4.4, how is the **without self-supervised** variant implemented? Does it refer to using the base model directly for inference without additional self-supervised training?

---

### Official Review · Reviewer_u436 · 2025-10-27

**Soundness:** 3
**Presentation:** 2
**Contribution:** 2
**Rating:** 4
**Confidence:** 4

**Summary:**

This paper introduces UniSplat, a self-supervised framework for learning unified 3D representations from unposed multi-view images. The method integrates three components: (1) a dual-masking strategy for geometry-aware learning, (2) a coarse-to-fine Gaussian splatting pipeline for refining appearance, and (3) a pose-conditioned recalibration mechanism for enforcing consistency between geometry and semantics.

The authors claim that UniSplat achieves robust 3D perception under unposed and sparse-view settings, generalizes across diverse tasks, and forms a “perceptual foundation for spatial intelligence.” Experiments are conducted on ScanNet, RealEstate10K, and a set of embodied AI benchmarks, showing improvements over prior pose-free baselines.

**Strengths:**

(1) The architecture combines several meaningful ideas — e.g., multi-level Gaussian refinement and multi-head consistency — that are intuitively complementary.

(2) The writing is clear, and the motivation for unified geometry–semantics–appearance modeling is well-articulated.

(3) Results on ScanNet indicate measurable gains in segmentation, depth, and rendering metrics over prior self-supervised methods.

(3) The idea of cross-task recalibration using estimated poses is interesting and could encourage better 3D–semantic alignment.

**Weaknesses:**

(1) **Overclaiming conceptual novelty:**
The paper frames UniSplat as a “unified self-supervised foundation for spatial intelligence”, which overstates its conceptual reach.
The proposed components (masking, Gaussian splatting, and cross-head consistency) are incremental combinations of existing trends — notably, dual-masking from VideoMAE [Wang et al. 2023], coarse-to-fine Gaussian strategies from Scaffold-GS [Lu et al. 2024], and geometry-semantic reprojection from prior multi-task 3D learning works.
The “unified” claim appears to stem from integrating these techniques rather than introducing a fundamentally new paradigm.

(2) **Ambiguity in “self-supervision”:**
The model relies heavily on distillation from pre-trained teachers (LSeg, VGGT) for both semantic and geometric priors.
This contradicts the “self-supervised” framing, as these teachers are trained with substantial supervision.
The resulting pipeline is more accurately pseudo-supervised or teacher-assisted pretraining, not purely self-supervised learning from raw unposed images.

(3) **Insufficient experimental validation:**
Ablations are limited to ScanNet and RealEstate10K, with no evaluation on out-of-distribution datasets (e.g., CO3D, KITTI, Replica, or ACID). This makes it difficult to assess the claimed “generalization across domains and tasks.”
The pose-conditioned recalibration mechanism is central to the paper, yet there is no quantitative evaluation or visualization of its contribution beyond a single ablation line in Table 3.
The paper does not show whether pose estimates are accurate, stable, or beneficial under large viewpoint shifts.

(4) **Lack of clarity on failure cases:**
The paper does not present qualitative examples where UniSplat fails or underperforms, which would be crucial for assessing its robustness to noise, textureless regions, or dynamic content.
The conclusion asserts that the model provides a “foundation for spatial intelligence,” yet there is no direct evidence of reasoning, planning, or long-horizon understanding tasks to support this statement.

**Questions:**

(1) **Clarify supervision hierarchy:**
To what extent are VGGT and LSeg required during training? Could UniSplat be trained without these external priors, and if so, how would performance degrade?

(2) **Pose estimation reliability:**
How accurate are the internally estimated poses? Have you compared them with ground truth on ScanNet to verify geometric plausibility?

(3) **Scope of generalization:**
Have you tested on outdoor or dynamic datasets to support the “spatial intelligence” claim? If not, the language should be toned down to “indoor scene understanding.”

---

### Official Review · Reviewer_GKfm · 2025-10-30

**Soundness:** 3
**Presentation:** 3
**Contribution:** 3
**Rating:** 6
**Confidence:** 2

**Summary:**

This paper proposes UniSplat, a framework for unified 3D scene understanding from unposed multi-view images. The key contributions include: (1) a geometry-aware dual-masking strategy to encourage 3D reasoning, (2) a hierarchical coarse-to-fine Gaussian splatting representation that refines the radiance field, (3) pose-conditioned recalibration to enforce cross-task geometric consistency

The method achieves strong pose-free quality for novel view synthesis without requiring SfM or per-scene optimization. Moreover, as a frozen encoder, it attains top performance across multiple embodied tasks.

**Strengths:**

1. The paper introduces three key innovations: dual masking, a coarse-to-fine Gaussian hierarchy, and pose-conditioned recalibration. These components work together to enable a pose-free, feed-forward approach to learning 3D geometry from unposed multi-view images.
2. Each module has a clear objective and is designed to contribute to the overall model. The method achieves strong results across ScanNet, RealEstate10K, and robotics benchmarks.
3. The paper is clearly written, with helpful figures and formulas. Training details and ablation analyses are provided, and sufficient comparison results are included as well.
4. The work offers an SfM-free and per-scene-optimization-free approach. It provides a framework for RGB-only 3D perception that benefits downstream 3D tasks while achieving strong performance.

**Weaknesses:**

1. Heavy reliance on teacher models: The proposed self-supervised method relies on LSeg and VGGT distillation rather than purely self-supervised. This may introduce teacher biases and limits the claim of self supervision.
2. Module efficiency vs. complexity: In ablation studies, dual masking and coarse-to-fine splatting bring only modest performance gains, yet introduce additional computational cost and system complexity.
3. Hyperparameter sensitivity: The method performance depends on mask ratios and gaussian latent tokens. The current exploration is limited, raising concerns about robustness cross datasets.

**Questions:**

1. What is the concrete advantage of distilling from LSeg/VGGT into UniSplat instead of directly using these SOTA models as the visual backbone?
2. Dual masking and coarse-to-fine splatting add computational overhead but provide modest gains. Can you provide a compute comparison: with and without these modules — to illustrate the trade-off between efficiency and performance?
3. How stable are the mask ratios and Gaussian token counts across different datasets? Can you provide variance analysis to demonstrate robustness of the current setting?

---

### Official Review · Reviewer_iuFM · 2025-11-02

**Soundness:** 3
**Presentation:** 3
**Contribution:** 2
**Rating:** 4
**Confidence:** 4

**Summary:**

This paper presents UniSplat, a self-supervised framework that learns 3D spatial representations from unposed multi-view images without requiring camera calibration or explicit geometric priors. The method introduces a unified latent space that jointly encodes scene appearance, depth, and spatial consistency using a multi-view feature alignment objective and a geometry-guided splatting decoder.

**Strengths:**

1, Learning 3D representations from unposed data is a genuinely challenging and valuable problem, given the increasing prevalence of Internet-scale uncalibrated imagery.

2, The framework is well-engineered, combining multi-view correspondence, depth estimation, and differentiable rendering into a cohesive pipeline.

3, The proposed recalibration module is a well-motivated and technically sound component that explicitly enforces geometric–semantic–appearance consistency within the predicted 3D scene. By re-projecting 3D point and semantic maps into the image plane using the estimated camera parameters, the mechanism provides a concrete spatial anchor that ties together outputs from different decoder heads. This design elegantly compensates for the lack of ground-truth camera supervision in unposed settings and introduces an implicit self-calibration loop that encourages all modalities to converge toward a coherent spatial frame.

4, UniSplat achieves reasonable results on benchmark datasets for both view synthesis and representation learning. Ablations show that each component contributes positively to performance, indicating a sound implementation.

**Weaknesses:**

1, The proposed “unified framework” mainly integrates well-known components: multi-view feature alignment (as in Spatial-ViT, MV-SSL), differentiable splatting/rendering (as in Gaussian Splatting, IBRNet), and self-distillation (as in DINO / iBOT). The central idea is a combination rather than a fundamentally new principle. The paper does not provide a new learning theory, loss formulation, or representation insight.

2, The paper assumes that multi-view consistency induces 3D awareness, but provides no analysis or quantification of learned geometry.

3, While the paper attributes the model’s improved geometry awareness to the proposed dual masking strategy, the mechanism lacks convincing theoretical or empirical justification.  Masking encoder and decoder tokens may regularize training and improve feature robustness, but it does not explicitly enforce cross-view geometric consistency or physical projection constraints—the key requirements for genuine 3D structure learning.  Moreover, the “geometry-guided” second-stage mask depends on coarse Gaussian fields that are themselves predicted from unposed inputs and thus may not provide reliable structural cues, creating a circular dependence.  Consequently, it remains unclear whether the observed performance gains stem from enhanced geometry reasoning or simply from stronger feature regularization.  Additional ablations or visualization of learned depth and correspondence would be necessary to substantiate the claimed geometric benefit.

**Questions:**

See the weakness part.

---

### Note · Authors · 2025-11-13

I have read and agree with the venue's withdrawal policy on behalf of myself and my co-authors.